# Co-Infection of SARS-CoV-2 and *Klebsiella pneumoniae*: A Systematic Review and Meta-Analysis

**DOI:** 10.3390/diagnostics14111149

**Published:** 2024-05-30

**Authors:** Angelica de Lima das Chagas, Joilma Cruz da Silva Araújo, Jaqueline Correia Pontes Serra, Kelliane Martins de Araújo, Marcos de Oliveira Cunha, Amanda dos Reis Correia, Laura Maria Barbosa Gonçalves, Lilian Carla Carneiro

**Affiliations:** 1Health Science Post Graduation, Federal University of Goias, Goiania 74605-050, Goias, Brazil; angelicalimac@discente.ufg.br (A.d.L.d.C.); jaquelinepontes@discente.ufg.br (J.C.P.S.); kellmsa@discente.ufg.br (K.M.d.A.); marcos.cunha@discente.ufg.br (M.d.O.C.); amanda_reis@discente.ufg.br (A.d.R.C.); 2Pharmacy Faculty, Federal University of Goias, Goiania 74605-170, Goias, Brazil; joilmacruz@discente.ufg.br; 3Postgraduate Program in Biology of Host Parasite Relationship, Federal University of Goias, Goiania 74690-900, Goias, Brazil; laura.barbosa@ufr.edu.br

**Keywords:** bacterium, epidemiology, virology

## Abstract

The study aimed to assess the prevalence of COVID-19 and *Klebsiella* spp. coinfection across continents. Conducted following PRISMA guidelines, a systematic review utilized PubMed, Embase, SCOPUS, ScienceDirect, and Web of Science databases, searching for literature in English published from December 2019 to December 2022, using specific Health Sciences descriptors. A total of 408 records were identified, but only 50 were eligible, and of these, only 33 were included. Thirty-three references were analyzed to evaluate the correlation between COVID-19 and *Klebsiella* spp. infections. The tabulated data represented a sample group of 8741 coinfected patients. The findings revealed notable disparities in co-infection rates across continents. In Asia, 23% of individuals were infected with *Klebsiella pneumoniae*, while in Europe, the proportion of co-infected patients stood at 15%. Strikingly, on the African continent, 43% were found to be infected with *Klebsiella pneumoniae*, highlighting significant regional variations. Overall, the proportion of *Klebsiella pneumoniae* co-infections among COVID-positive individuals were determined to be 19%. Particularly concerning was the observation that 1 in 6 ICU coinfections was attributed to *Klebsiella pneumoniae*, indicating its substantial impact on patient outcomes and healthcare burden. The study underscores the alarming prevalence of co-infection between COVID-19 and *Klebsiella pneumoniae*, potentially exacerbating the clinical severity of patients and posing challenges to treatment strategies. These findings emphasize the importance of vigilant surveillance and targeted interventions to mitigate the adverse effects of bacterial coinfections in the context of the COVID-19 pandemic.

## 1. Introduction

In December of the 2019 year, China confirmed to the World Health Organization (WHO) the events of abnormal cases of pneumonia in Wuhan, Hubei region. In the first week of January, the Chinese authorities confirmed that it was a virus, the coronavirus, then called severe acute respiratory syndrome coronavirus-2 (SARS-CoV-2), the etiological agent of the disease COVID-19 [1]. It is an acute infection with a high chance of transmission. This viral type favored a rapid transmission process, and then community transmission was revealed, leaving Asia in a short time and reaching Europe, Oceania, and finally America.

The WHO decreed the pandemic on 11 March 2020, with more than 100,000 cases spread across 114 countries, and on 20 March, just 9 days later, community transmission of the virus in Brazil was confirmed [2]. Through the discussion forum of the organization of the virological society, the results of the first complete genome sequencing of SARS-CoV-2 in Brazil were published, which was carried out 48 h after the confirmation of the first cases of COVID-19 in the country [3]. The first case was confirmed at the Lutz Institute by a resident of São Paulo who had just returned from a trip to the Lombardy region of Italy. Due to the rapid geographic spread of the new coronavirus, on 11 March 2020. The WHO characterized COVID-19 as a pandemic caused by SARS-CoV-2 [4].

COVID-19 has a wide spectrum of presentation, with respiratory problems, fever, and sore throats predominating; it can even remain asymptomatic. According to the WHO [4], 80% of patients with COVID-19 have mild symptoms and no complications, 15% progress to hospitalization requiring oxygen therapy, and 5% need to be treated in an Intensive Care Unit (ICU). According to Yusof et al. [5], the ICU is a specific place to receive critically ill patients where invasive procedures are performed, such as the insertion of a central venous catheter, probes, or orotracheal intubation, among others. Patients who are hospitalized in these health environments are at high risk of acquiring infections, as well as increasing the chances of colonization by multidrug-resistant microorganisms.

Healthcare-associated infections (HAIs) are one of the main complications of the clinical picture of hospitalized patients, affecting thousands of people worldwide. Infections increase morbidity, mortality among patients, and costs associated with health care [6]. In these infections, there is a growing manifestation of bacterial isolates that do not respond to antimicrobial therapy and are therefore considered resistant. Infections with treatment-resistant bacteria have clinical manifestations similar to those originating from susceptible organisms. However, treatment alternatives become greatly reduced in the presence of resistant organisms [7,8,9].

The prevalence of resistant bacterial strains varies according to the health establishment, the specialty, geographic location, length of stay of the patient, and his profile in the service [6,9,10].

The prevention and treatment of infectious diseases is related to reducing morbidity and mortality and increasing life expectancy [11]. These factors are intrinsically associated with our better understanding of the molecular mechanisms of these diseases, as well as the interest of pharmaceutical companies in the development of safer and more active antimicrobial compounds, such as the well-known beta-lactams and phenylpropanoids [12].

Antibiotic resistance can happen naturally from the bacterial ability to adapt. The indiscriminate use of antimicrobials allows greater exposure to bacteria and provides opportunities for the acquisition of resistance mechanisms. Antimicrobial resistance has become a public health problem worldwide. High levels of bacterial resistance to gentamicin, imipenem, meropenem, cefepime, ciprofloxacin, levofloxacin, aztreonam, and piperacillin + tazobactam were observed [13].

Studies show that COVID-19 may be associated with bacterial coinfections. Among the contagious pathogens found is *Klebsiella pneumoniae* [14]. *Klebsiella pneumoniae* is a Gram-negative, encapsulated bacterium; its infection is caused by strains called “classic” *K. pneumoniae* (cKp), in which it colonizes the gastrointestinal tract as well as the nasopharynx. Still, these strains persist in hospital settings and cause infections in debilitated patients [15]. *K. pneumoniae* stands out for its ability to develop enzymatic resistance mechanisms and for the creation of a multidrug resistance phenotype to the exacerbated use of antimicrobials [8].

Multidrug-resistant microorganisms (MDR) are those resistant to three or more classes of antimicrobials, the most relevant being *Pseudomonas aeruginosa*, *Klebisiella pneumoniae*, *Acinetobacter baumannii*, *Staphylococcus aureus*, and *Enterococcus faecium*, which are associated with colonization/infection [16,17]. Colonization is defined, according to Anvisa, as the presence of microorganisms on the skin, mucous membranes, exposed lesions, detected through laboratory cultures; however, without causing an immune response to the individual, without changes in the normal functions of the organ. Infection with microorganisms causes organic changes, which appear through signs or symptoms [7].

The KPC bacteria (*Klebsiella pneumoniae* Carbapenemase), the “superbug”, was identified for the first time in the United States in 2000 after suffering a genetic mutation that gave it resistance to multiple antibiotics (carbapenems, in particular) and the ability to make other bacteria resistant. It can also cause pneumonia, blood infections, in the urinary tract and surgical wounds, and illnesses that can evolve into a generalized infection, often fatal [18].

Bacterial resistance is a public health problem, impacting the economy and health security at an international level. Zhou et al. [19] estimate that the increase in bacterial resistance will be related to 10 million deaths per year by 2050. In Uganda, relevant advances were made by the national antimicrobial resistance (AMR) program guided by the WHO Global Surveillance System [20]. With WHONET 5.5 software, it is possible to manage and monitor ARM data [21], In 2019, six main pathogens related to bacterial resistance and death were identified: *Escherichia coli*, followed by *Staphylococcus aureus*, *Klebsiella pneumoniae*, *Streptococcus pneumoniae*, *Acinetobacter baumannii*, and *Pseudomonas aeruginosa*, which were implicated in 3.57 million ARM-related deaths [8,22]. *S. aureus*, *S. pneumoniae*, *Streptococcus pyogenes*, and *Haemophilus influenzae* were involved in secondary infections [11,23,24,25,26,27,28,29].

In this sense, the focus of the present study is to contribute significantly to the literature, taking into account the little knowledge In this sense, the focus of the present study is to contribute significantly to the literature, taking into account the little knowledge in the world literature on the syndrome and clinical evolution of COVID-19 in co-infection with *Klebsiella* spp. As well as, at the same time, contributing to the outline of the epidemiological profile with SARG and the incidence of COVID-19. In addition, the study of co-infection with *Klebsiella pneumoniae*, associated with COVID-19, will help to carry out the antimicrobial profile and thus may help in the effectiveness and speed of treatment, providing a study that will contribute to a more efficient treatment.

There is much uncertainty about the impact of bacterial co-infections during the pandemic, especially in tensive care units, that needs to be assessed from a global health perspective. There is considerable uncertainty surrounding the impact of bacterial co-infections during the pandemic, particularly in intensive care units, necessitating assessment from a global health perspective. As described by Brogna et al., the clinical importance of co-infections caused by the SARS-CoV-2 coronavirus must be emphasized, given its bacteriophage behavior and the ability to infect human bacterial flora [30].

The aim of this work was to determine the prevalence of bacterial co-infection in patients hospitalized due to COVID-19 and to correlate it with *Klebsiella* spp. at the same time as the virus.

## 2. Materials and Methods

This systematic review examined hospitalized patients with COVID-19 and *Klebsiella* spp. A review was performed based on PRISMA (Figure 1) guidelines to determine the rates of concomitant infections in patients.

### 2.1. Information Sources

The meta-analysis utilized databases including PubMed, Embase, Scopus, and ScienceDirect, spanning from December 2019 to December 2022. Employing descriptors such as “Klebsiella”, “bacterial infection”, “Coronavirus”, “SARS-CoV-2”, “COVID-19”, “co-infection”, and “transversal”, in conjunction with Boolean operators AND and OR, yielded a comprehensive dataset.

A total of 184 references were retrieved from PubMed, 75 from Embase, 87 from Scopus, 28 from ScienceDirect, and 34 from Web of Science, culminating in 408 articles. After removing 42 duplicates and excluding 375 articles that did not meet inclusion criteria, 33 studies were deemed suitable for systematic review and meta-analysis, analyzed according to the PRISMA protocol for systematic review and meta-analysis [17].

The data evaluation process was conducted by two independent statistical reviewers to ensure rigorous analysis and interpretation of the findings. This meticulous approach aimed to enhance the reliability and validity of the study outcomes, facilitating robust conclusions regarding the prevalence and impact of co-infections between *Klebsiella* spp. and COVID-19.

### 2.2. Eligibility Criteria

The study employed specific inclusion and exclusion criteria to ensure the selection of relevant literature. Inclusion criteria encompassed prospective, cross-sectional, and retrospective studies providing evidence of co-infection between Klebsiella and SARS-CoV-2, supported by laboratory confirmation, focusing on patients admitted to intensive care units (ICUs).

Conversely, studies failing to address co-infection cases between Klebsiella and SARS-CoV-2, as well as those involving animal subjects, conference abstracts, letters, editorials, reviews, case reports, and randomized controlled trials, were excluded from consideration. Any discrepancies or uncertainties regarding the inclusion or exclusion of studies were resolved through consultation with an independent reviewer not involved in the initial screening process. This rigorous approach aimed to uphold the integrity and validity of the study by ensuring the selection of high-quality, relevant literature that aligns with the research objectives.

### 2.3. Data Extraction

Information was collected according to the variables: author, year of publication, country, and study design. The extraction was carried out by independent authors (Table 1).

### 2.4. Data Analysis

In assessing co-infection by *K. pneumoniae*, the study employed a proportion test with a significance level set at 5%. To ensure a comprehensive analysis, both randomized and fixed effects were considered, accounting for heterogeneity among the selected studies. The mixed generalized linear method (GLMM) with log transformation (PLOGIT) was utilized, enhancing result accuracy. The use of this statistical approach contributed to the robustness and reliability of the findings. All analyses were conducted using R Studio^®^ 4.02 [59], a widely recognized software known for its versatility and capability in statistical modeling and data analysis. By employing these rigorous analytical methods and software, the study aimed to provide precise and informative insights into the prevalence and correlation of co-infection with *K. pneumoniae* in COVID-19 patients, thereby contributing to a better understanding of the disease dynamics and potential treatment strategies.

## 3. Results

### 3.1. Result of the Study Identification Process

Upon conducting a comprehensive search of the databases, a total of 408 surveys were initially identified using the specified descriptors. Following the elimination of duplicate records, 366 studies underwent screening based on their titles and abstracts. Subsequently, 50 studies were selected for a thorough examination of their full texts. After careful assessment, 33 studies were deemed eligible and were included in the review (Figure 1). This systematic approach to study selection ensured the inclusion of relevant and pertinent research while excluding duplicates and studies that did not meet the predetermined criteria. By meticulously filtering the literature in this manner, the review maintained a high standard of quality and rigor, thereby enhancing the reliability and credibility of the findings (Figure 1).

### 3.2. Analysis of Extracted Data

Across the Asian continent, a comprehensive investigation involved 22 surveys, collectively involving 2627 participants. The analysis revealed a notable degree of heterogeneity among the surveys, with an I² value of 94%, suggesting substantial variability in the findings. Despite this heterogeneity, the proportion of individuals co-infected with *K. pneumoniae* was determined to be 23%. This proportion, represented by a confidence interval of 0.14 to 0.35, underscores the substantial presence of co-infection with *K. pneumoniae* among COVID-19 patients in the Asian region (Figure 2).

In Europe, a total of 10 surveys involving 3228 participants were selected for analysis. The heterogeneity among these surveys was substantial, with an I^2^ value of 95%, indicating significant variability in the findings. Despite this variability, the proportion of individuals co-infected with *K. pneumoniae* was determined to be 15%. This proportion, represented by a confidence interval of 0.06 to 0.32, underscores the prevalence of co-infection with *K. pneumoniae* among COVID-19 patients in the European region. However, the wide confidence interval suggests some uncertainty in the precise estimate, highlighting the need for further research to refine our understanding of co-infection patterns in this region (Figure 3).

Conversely, only one survey was conducted in Africa, with 28 patients, revealing a notably high proportion of individuals co-infected with *K. pneumoniae*, at 43%. Although this finding is based on a single survey and should be interpreted with caution, it suggests a potentially significant burden of co-infection on the African continent, warranting further investigation.

Similarly, on the American continent, two surveys involving 2858 participants were combined for analysis. The proportion of individuals co-infected with *K. pneumoniae* was found to be 4%, with a confidence interval of 0.04 to 0.05. While this proportion is relatively lower compared to other continents, the presence of co-infection highlights the importance of considering bacterial pathogens alongside COVID-19 in clinical management and public health strategies.

Overall, these findings underscore the global variability in the prevalence of co-infection with *K. pneumoniae* among COVID-19 patients. The heterogeneity observed across continents highlights the need for region-specific approaches to understanding and addressing co-infection patterns, informing targeted interventions, and healthcare resource allocation to effectively manage the dual burden of viral and bacterial infections. (Figure 3).

A comprehensive statistical analysis encompassing all studies was conducted to assess the proportion of *Klebsiella pneumoniae* infections within the population that was co-infected with bacteria in general (*n* = 8741). The analysis revealed that 19% [13% to 28%] of individuals in this population were infected with *Klebsiella pneumoniae*, translating to approximately 1422 patients co-infected with this bacterium. Remarkably, this finding indicates that 1 out of every 6 co-infections in intensive care units (ICUs) were attributed to *K. pneumoniae* (Figure 4).

This significant proportion underscores the substantial impact of *K. pneumoniae* co-infections among COVID-19 patients, particularly in ICU settings. The prevalence of *K. pneumoniae* as a co-infecting pathogen highlights its clinical significance and the need for targeted management strategies to address bacterial co-infections alongside COVID-19.

## 4. Discussion

The discussion surrounding coinfections in the context of COVID-19 is complex and multifaceted, encompassing various factors such as pathophysiology, epidemiology, clinical management, and public health implications. Understanding the interplay between SARS-CoV-2 and bacterial pathogens like *Klebsiella pneumoniae* is crucial for optimizing patient care and mitigating the impact of coinfections on disease outcomes [60,61].

The increasing prevalence and variability of severe acute respiratory syndrome coronavirus 2 (SARS-CoV-2) have resulted in significant deficiencies in the pathogenic mechanisms, and last resort control relies on supportive care. Thus, in the absence of widely available specific therapies, most efforts rely on supportive care, biocontainment protocols, and mitigating the role of co-infections that exacerbate the disease. The role of co-infections in SARS-1 and Middle East respiratory syndrome (MERS) outbreaks has been well characterized [60,61,62].

The epidemiology of viral-bacterial coinfections in COVID-19 exhibits temporal and geographical variability, influenced by factors such as population demographics, healthcare infrastructure, infection control measures, and antimicrobial stewardship practices. Early reports from the pandemic’s onset highlighted the prevalence of bacterial coinfections in hospitalized COVID-19 patients, particularly in regions with high disease burden and healthcare-associated transmission [29,33,43].

Subsequent studies have provided insights into the evolving epidemiology of coinfections over time, with fluctuating rates observed across different phases of the pandemic. While initial waves may have been characterized by higher rates of coinfections due to overwhelmed healthcare systems and nosocomial transmission, subsequent phases with improved infection control measures and vaccination efforts may have led to reduced rates of bacterial coinfection [29,31,32,33,34,35,36,37,38,39,40,41,42,43,44,45,46,47,48,49,50,51,52,53,54,55].

In the first year of the pandemic, four reports worldwide indicated 13% of co-infections caused by SARS-CoV-2 and *Klebsiella pneumoniae* [26,43,49,58]. In the second year of the pandemic, 13 publications reported 11% of coinfections caused by *Klebsiella pneumoniae* [29,31,32,35,39,40,44,47,51,52,54,56,57].

In 2022, 17 studies reported 21% co-infection with SARS-CoV-2 and *Klebsiella pneumoniae* [23,25,28,33,34,36,37,38,41,42,45,46,48,50,53,55,63].

Study populations ranged from 18 to 1450 cases; the three countries that published the highest co-infection data were China, Iran, and Spain. Presented 42% of coinfections with *Klebsiella pneumoniae* in China (152/363), 22% in India (95/426), 19% in Iran (152/809) and 9% in Spain (15/166). Similar findings during the early stages of COVID-19 in Wuhan reported a 16% rate of secondary infections in hospitalized patients, which was higher among non-survivors than survivors (50% vs. 1%) [62,64].

The last two groups in the aforementioned study showed significant differences in the leukocyte count and absolute values of lymphocytes. In Spain, only 7.2% of coinfections (71/989) were reported during hospitalization, inconsistent with higher prevalence rates in other viral pandemics [35,62].

The need for combined therapy to treat coinfection processes was based on assumptions of previous pandemics, such as influenza, which resulted in poor human prognosis [60]. The relatively lower levels and variability in reported co-infections in SARS-CoV-2, compared to other pandemics, have hampered a universal consensus on co-infections [62].

In the United Kingdom (UK), for example, in a total of 836 patients, only 3.2% had an early co-infection and *S. aureus* was the most common pathogen [65]. In the present study, in 2022, epidemiological data from the UK showed that in a total of 254 patients, 33% had co-infection/co-colonization from admission to the end of their ICU stay, and 28% had co-infection with *Klebsiella pneumoniae*. Similarly, in France, 28% of bacterial co-infections in patients with severe SARS-CoV-2 pneumonia in the intensive care unit were mainly *S. aureaus*, Hemophilus influenzae, *S. pneumonia,* and Enterobacteriaceae [62,66].

Patients with coinfections had worse outcomes, and overall mortality was 9.8% [35]. Due to a lack of knowledge about co-infection, many decisions about antibiotic therapy for patients with COVID-19 were made based on clinical experience and limited scientific evidence [62].

It is noted that with the advancement of the pandemic and the passing of the years, there was a significant increase in viral x bacterial co-infections, which suggests difficulty in the treatment. Still, the studies used in the PRISMA model were predominantly based on patients for whom it was possible to collect samples and visualize bacterial co-infections. Consequently, the clinical picture may be underestimated at a global level, especially for individuals who underwent home care and did not have data collected. On the other hand, the vaccine significantly helped to prevent the clinical condition of the co-infected patient from worsening.

It is plausible that heightened immune reactions and hyperactivation of macrophages may have played a role beyond the empirical use of antibiotics and quarantine, which limit exposure [62,67].

Future studies will be needed to elucidate the role of factors involved in reducing superinfections. As extensively discussed by some authors, the bacteriophage behavior of the SARS-CoV-2 coronavirus, which also infects human bacterial flora, is an aspect of great relevance [30,68,69]. Genomic assessments of bacteria co-infected with the SARS-CoV-2 coronavirus constitute a promising field of research.

## 5. Conclusions

In this study, SARS-CoV-2 co-infection with *K. pneumoniae* was evaluated according to continent. It was observed that on the Asian continent, the published studies disclosed 23% of co-infection with *K. pneumoniae* with a ratio of 0.23 [0.14 to 0.35]. On the European continent, 15% of viral co-infection with *K. pneumoniae*, proportion 0.15 [0.06 to 0.32] have been published. On the American continent, 4% of cases of co-infection were published, with a proportion of 0.04 [0.04 to 0.05]. According to the analyses carried out in this study, it was found that the highest percentages of viral co-infection with *K. pneumoniae* occurred on the Asian continent. The results found in this study may have been influenced by the number of publications included on each continent.

Overall, the proportion of coinfection between positive COVID-19 patients and *K. pneumoniae* was 19%, which is considered a high rate of coinfection. When evaluating all continents together, for every six COVID-positive patients, one of them was simultaneously co-infected with *Klebsiella pneumoniae*.

The data are relevant since *Klebsiella* spp. has a history of resistance to antimicrobials, which is often not identified and which can complicate COVID-19. In this way, it is believed that there may be a contribution to COVID-19 treatments, making healthcare less expensive for patients and healthcare organizations.

Further research is needed to elucidate the underlying mechanisms driving the dynamics of viral-bacterial coinfections in COVID-19 and their implications for disease progression and treatment outcomes. Studies investigating host immune responses, microbial interactions, and the role of antimicrobial therapy in modulating coinfection outcomes are warranted. Additionally, efforts to optimize diagnostic strategies, refine treatment algorithms, and implement targeted interventions to prevent and manage bacterial coinfections in COVID-19 patients are essential for improving clinical outcomes and reducing healthcare-associated burdens.

In conclusion, the prevalence of coinfections between SARS-CoV-2 and *Klebsiella pneumoniae* remains significant, highlighting the importance of integrated approaches to managing COVID-19 patients. Heightened awareness, surveillance, and evidence-based interventions are crucial for mitigating the impact of bacterial coinfections on COVID-19 outcomes and reducing the overall burden of disease. Collaborative efforts between healthcare providers, researchers, policymakers, and public health authorities are essential for addressing the complex challenges posed by viral-bacterial coinfections in the context of the ongoing COVID-19 pandemic.

## Figures and Tables

**Figure 1 diagnostics-14-01149-f001:**
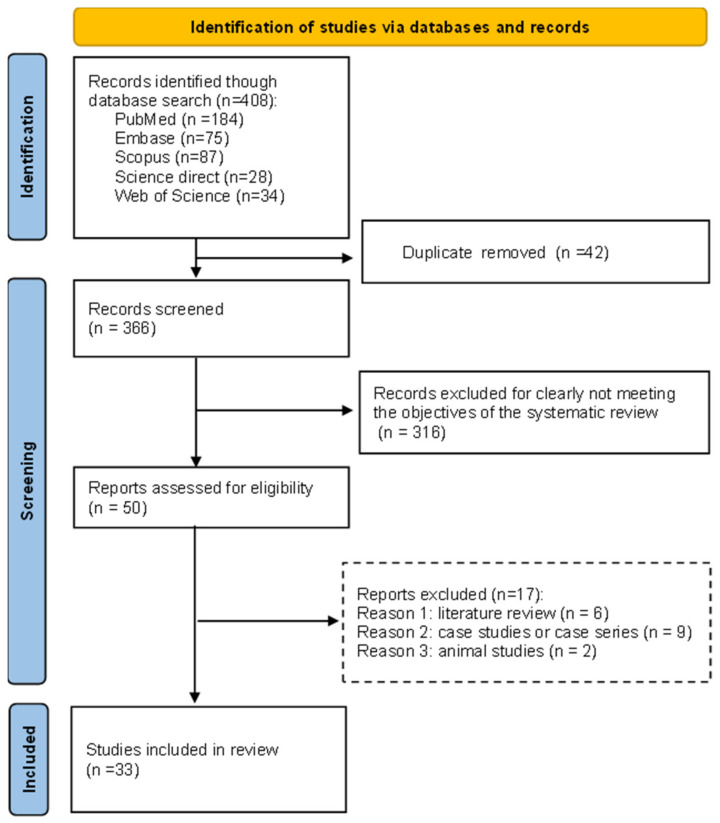
Organization chart of the sealing scheme at each stage of the process.

**Figure 2 diagnostics-14-01149-f002:**
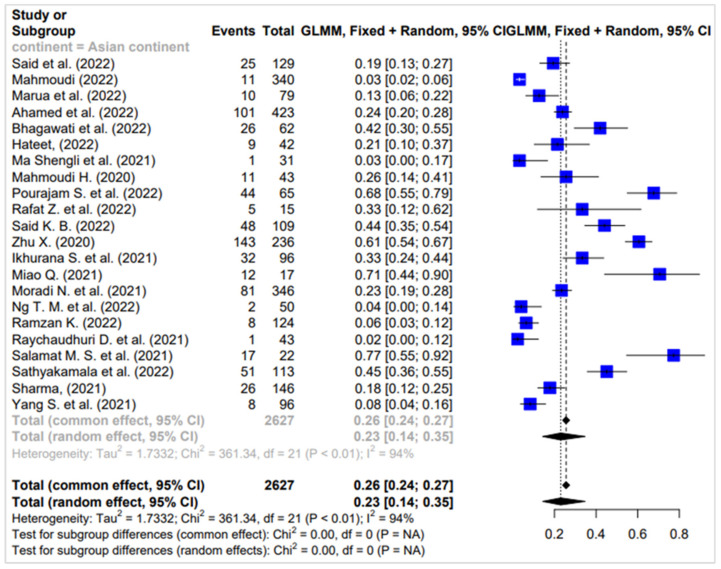
Proportion of co-infected people in surveys carried out on the Asian continent [23,26,28,29,32,34,37,39,40,42,44,46,48,50,51,53,54,55,57,58].

**Figure 3 diagnostics-14-01149-f003:**
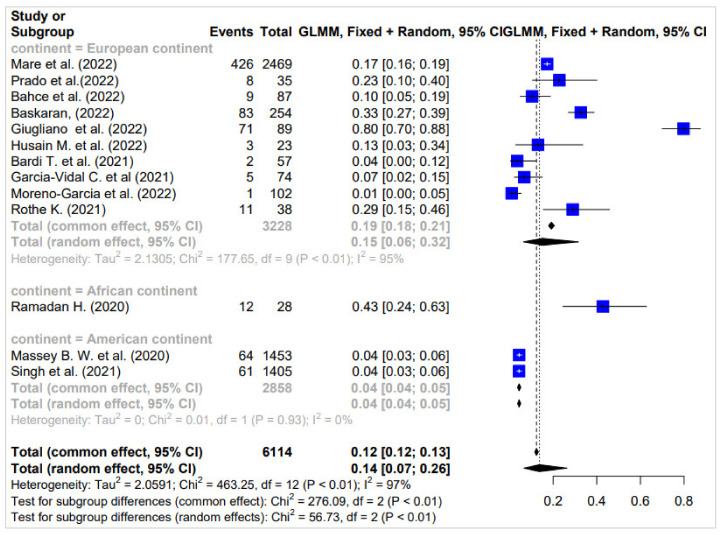
Proportion of co-infected people in surveys carried out on the European, African, and American continents [25,31,33,35,36,38,41,43,45,47,49,52,56].

**Figure 4 diagnostics-14-01149-f004:**
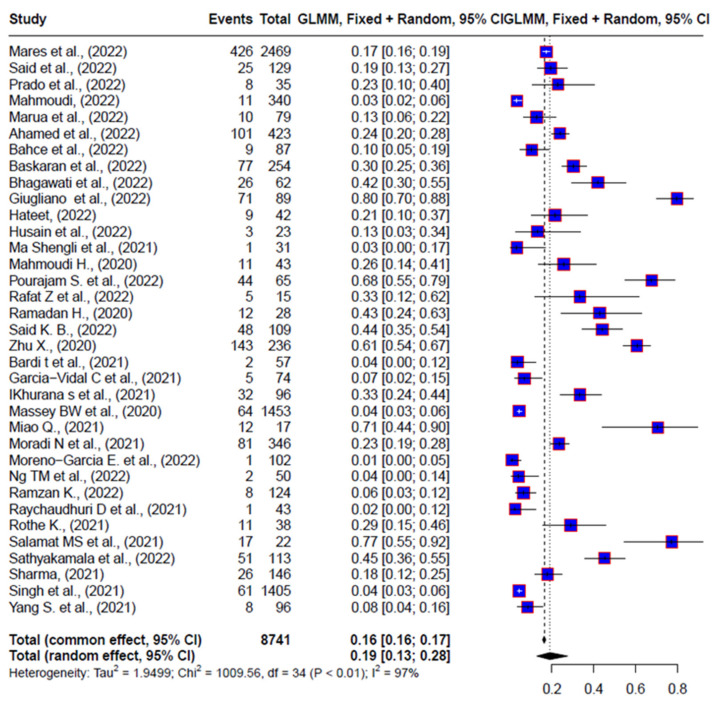
Proportion of *K. pneumoniae* infections, in a COVID-positive population co-infected with bacteria in general [23,25,26,28,31,32,33,34,35,36,37,38,39,40,41,42,43,44,45,46,47,48,49,50,51,52,53,54,55,56,57,58,59].

**Table 1 diagnostics-14-01149-t001:** Selected studies.

Author	Year	Country	Overall Desing
Bahçe et al. [25]	2022	Türkiye	prospective study
Bardi et al. [31]	2021	Spain	retrospective study
Moradi et al. [32]	2021	Iran	cross-sectional study
Ahamed et al. [23]	2022	Pakistan	cross-sectional study
Baskaran et al. [33]	2022	England	retrospective cohort study
Bhagawati et al. [34]	2022	India	retrospective study
Garcia-Vidal [35]	2021	Spain	observational cohort study
Giugliano et al. [36]	2022	Italy	cross-sectional study
Hateet [37]	2022	Iraq	prospective study
Husain et al. [38]	2022	France	retrospective observational study
Khurana et al. [39]	2021	India	cross-sectional study
Ma Shengli et al. [40]	2021	China	Cohort study
Mahmoudi [26]	2020	Iran	cross-sectional study
Mareș et al. [41]	2022	Romania	cross-sectional retrospective study
Marua et al. [42]	2022	India	cross-sectional study
Massey et al. [43]	2020	EUA	prospective study
Miao et al. [44]	2021	China	retrospective study
Moreno-García et al. [45]	2022	Spain	observational cohort study
Ng et al. [46]	2022	Singapore	observational cohort study
Pourajam et al. [28]	2022	Iran	retrospective cohort study
Prado et al. [47]	2022	Spain	cross-sectional study
Rafat et al. [48]	2022	Iran	cross-sectional study
Ramadan et al. [49]	2020	Egypt	prospective study
Ramzan et al. [50]	2022	Pakistan	retrospective study
Raychaudhuri et al. [51]	2021	India	prospective observational study
Rothe et al. [52]	2021	Germany	retrospective cohort study
Said et al. [53]	2022	Saudi Arabia	retrospective cross-sectional study
Salamat et al. [54]	2021	Philippines	retrospective study
Sathyakamala et al. [55]	2022	India	retrospective observational study
Sharma et al. [29]	2021	India	prospective and observational study
Singh et al. [56]	2021	EUA	retrospective cohort study
Yang et al. [57]	2021	China	retrospective observational study
Zhu et al. [58]	2020	China	retrospective study

## Data Availability

The data are included in the article.

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
