# Peer review of "Co-Infection of SARS-CoV-2 and Klebsiella pneumoniae: A Systematic Review and Meta-Analysis"

_diagnostics, 2024, doi:10.3390/diagnostics14111149_

Round 1
Reviewer 1 Report
Comments and Suggestions for Authors
Chagas et al. performed a systematic review and meta-analysis following PRISMA guidelines to evaluate SARS-CoV-2 and Klebsiella spp. coinfections. Case studies were divided on the basis on continents to understand regional and healthcare variations. The article brings the public focus on the importance of monitoring co-infections and their impact on disease outcomes. This must be taken into focus during treatment as well. Here are some suggestions for the article:
Line 15: expand PRISMA
Line 17 “Thirty-five references”: The method says 33. Kindly recheck.
Line 36: Replace “flying mammals” with name of animals where coronavirus was found (bats, camels, etc)
Line 37-38: The sentence id confusing, kindly rephrase.
Line 119: I don’t believe based on the number of literatures that “little knowledge” is the correct phrase to be used here. Kindly reword.
Line 132: Expand PRISMA again.
Line 150-166: This section can be merged with the previous section as it is already said above that 33 studies are finalized based on inclusion criteria.
Figure 2, 3, 4: Align headings with the columns better. Label x-axis.
Line 217: Add total number of participants.
Reviewer 2 Report
Comments and Suggestions for Authors
The authors conducted a review study using the Prisma method, and observed a correlation between Klebsiella bacterial co-infections and SARS-CoV-2. Although well structured, the study needs improvement.
The authors need to:
1. better point out in the discussion that the studies used for the Prisma ù are those mainly that described in patients for whom sample collection and visualization of bacterial co-infection was possible, and that therefore the clinical picture could be underestimated for those in the world who performed home care and did not collect data.
2. The topic of bacterial cofactor and SARS-CoV-2 has been discussed extensively by some Italian authors who have observed the bacteriophage behavior of the virus, it would be indicated to refer to these papers as well in both the introduction and discussion.
3. The authors should add the completed prisma model (https://www.prisma-statement.org/ )in the supplementary materials.
Round 2
Reviewer 2 Report
Comments and Suggestions for Authors
The authors resolved the requests. Only at line 134 the name of the cited study does not match with the authors of reference 30.
Author Response
The point raised in line 134 about reference 30 was corrected.
"As described by Brogna et al..."